# Revealing topology in metals using experimental protocols inspired by *K*-theory

Wenting Cheng [1,5] ✉, Alexander Cerjan [2,5] ✉, Ssu-Ying Chen[1] ✉, Emil Prodan [3] ✉, Terry A. Loring[4] ✉ & Camelia Prodan[1] ✉

Topological metals are conducting materials with gapless band structures and nontrivial edge-localized resonances. Their discovery has proven elusive because traditional topological classification methods require band gaps to define topological robustness. Inspired by recent theoretical developments that leverage techniques from the field of $C^*$-algebras to identify topological metals, here, we directly observe topological phenomena in gapless acoustic crystals and realize a general experimental technique to demonstrate their topology. Specifically, we not only observe robust boundary-localized states in a topological acoustic metal, but also re-interpret a composite operator—mathematically derived from the *K*-theory of the problem—as a new Hamiltonian whose physical implementation allows us to directly observe a topological spectral flow and measure the topological invariants. Our observations and experimental protocols may offer insights for discovering topological behaviour across a wide array of artificial and natural materials that lack bulk band gaps.

Over the past two decades, immense progress has been made in predicting and observing topological phases of matter and their associated boundary-localized states in insulators[1–5] and semimetals[6–18]. These developments have been predicated upon the spectral isolation of the topological phenomena in these classes of systems; although materials such as Dirac semimetals[6–8], Weyl semimetals[9–15], and nodal line/ring semimetals[16–18] generally do not possess complete gaps in their band structures, the topological phenomena that manifest in these systems nevertheless appear within incomplete band gaps, allowing their boundary-localized states to be uniquely identified at some energy and quasimomentum. In contrast, the lack of band gaps (or, more generally, mobility gaps) in metals and other types of gapless systems has made their topological analysis extremely challenging and presently there are concentrated efforts in this direction[4]. While previous works have studied the bulk properties of topological metals, such as their topological responses and their relation to the geometry and topology of the Fermi surface[19–23], the focus of our work is on topological bulk-boundary correspondence in metals. As such, for the purposes of this study, we are using 'topological metal' to specifically refer to systems that exhibit a bulk-boundary correspondence that can be predicted using an invariant determined in the system's bulk, but whose topologically protected boundary-localized states or resonances are always degenerate in both energy and wave vector with bulk states. (In contrast, topological states in insulators and semimetals exhibit a range of energies and wave vectors where no bulk states exist). From a theoretical perspective, the absence of spectral or dynamical gaps essentially precludes the use of topological band theory to identify the invariants of these systems, and prohibits such theories from predicting a measure of topological protection. Moreover, any boundary-localized phenomena in gapless systems will generally hybridize with the degenerate bulk states to create boundary-localized resonances, which complicates their experimental observation. Thus, despite the enormous advances that

[1]Department of Physics, New Jersey Institute of Technology, Newark, NJ, USA. [2]Center for Integrated Nanotechnologies, Sandia National Laboratories, Albuquerque, NM 87185, USA. [3]Department of Physics, Yeshiva University, New York, NY, USA. [4]Department of Mathematics and Statistics, University of New Mexico, Albuquerque, NM 87131, USA. [5]These authors contributed equally: Wenting Cheng, Alexander Cerjan. ✉e-mail: wc327@njit.edu; awcerja@sandia.gov; sc945@njit.edu; prodan@yu.edu; loring@math.unm.edu; cprodan@njit.edu

have been made in topological materials, the study of topological metals has remained almost entirely unexplored.

Recently, a general theoretical method for evaluating the topology of metallic and gapless systems was put forward, opening new opportunities for discovering topology in this class of systems that could not be previously explored[24]. This theoretical framework is rooted in the system's spectral localizer, which makes use of the system's real-space description and yields local invariants (synonymous with local markers) that are protected by local gaps[25–27]. The key concept that links traditional band theoretic approaches to this local understanding of a system's topology stems from a dual description of atomic limits, i.e., the limit where a complete basis of spatially localized Wannier functions exists. In band theoretic approaches, a group of bands is topologically trivial if they can be continued to an atomic limit without closing the band gap or breaking a symmetry; any obstruction to this continuation manifests as a non-trivial invariant[28–31]. From a real-space operator perspective, an atomic limit's complete set of Wannier functions each have both a well-defined position and energy (in crystals they can be expressed as a flat band[32]). Thus, a $d$-dimensional atomic limit's Hamiltonian, $H^{(AL)}$, commutes with its position operators, $X_j^{(AL)}$, $[H^{(AL)}, X_j^{(AL)}] = 0$, for all $j \in 1, \ldots, d$. Using this mathematical observation, the spectral localizer ascertains a system's topology by determining whether there is an obstruction in continuing its $H$ and $X_j$ to be commuting (given similar restrictions as before), an analysis that can be performed using recent developments from the study of $C^*$-algebras[25,33]; any obstruction to continuing the system to possess commuting matrices yields a non-trivial invariant.

If a system's topology is instead linked to its multiple inequivalent atomic limits (e.g. as is common in systems with chiral- or crystalline-based topology), a system is only considered trivial if it can be continued to a chosen trivial atomic limit. Such cases are automatically handled by the spectral localizer, where it is necessary to explicitly specify the system's boundary and the grading operator that defines the chiral or crystalline symmetry to evaluate the possibility of continuation. This pair of choices effectively fixes which atomic limit is considered to be trivial, and the spectral localizer ascertains whether there is an obstruction to continuing a given system to this trivial atomic limit. Altogether, by recasting the determination of a system's topology to real space, and not relying upon a bulk band gap to be the measure of topological protection, the spectral localizer is equally applicable to insulators and metals—meaningful local gaps that protect non-trivial topology can be found in both cases[24]. However, despite the prediction that the local topological invariants of gapless systems can be robust against system perturbations, robust topological metals that possess a bulk-boundary correspondence have not been previously identified in any platform.

Here, we theoretically develop and experimentally realize robust boundary-localized states protected by a bulk topological invariant in a gapless acoustic crystal. Unlike the forms of topology that can be found in semi-metals, the topological states we observe are degenerate with bulk states in both energy and wave vector. Our design is based on coupling a topologically gapped acoustic crystal to a gapless one, yielding a system that full-wave simulations show possesses a gapless resonant spectrum. Nevertheless, when domain boundaries are introduced, both simulations and experimental observations reveal that this two-layer system possesses boundary- and domain-localized states, and the topological origins of these states can be proven using the spectral localizer. To confirm the topological origin of these localized states, we develop an experimental protocol that treats the system's spectral localizer as the Hamiltonian of a related system, enabling the direct observation of the underlying system's spatially resolved $K$-theory, i.e., its local topology. Taken together, these measurements demonstrate that

we have realized a topological metal. Given the generality of our experimental methodologies, these findings open opportunities to discover gapless topological phenomena across a broad range of natural and artificial materials.

## Results

### Gapless topological phononic crystal

To realize our proof-of-principle topological metal, we use a phononic crystal, as this platform has been proven to be straightforward to fabricate and reconfigure (see Fig. 1a, b)[12–15,34–47]. Heuristically, our aim for demonstrating such a phononic topological metal is to start with a topological insulator, couple it to a second lattice such that the combined system is gapless, and then probe it to observe boundary-localized states. For our specific design, we take the initial topological insulator to be a Su–Schrieffer–Heeger (SSH) lattice[48], whose gapped spectrum is shown in Fig. 1c. The second lattice is chosen to be a 1D monatomic lattice with uniform couplings, whose gapless spectrum is shown in Fig. 1d. Here, we are enforcing the resonator geometry and spacing to be the same for both lattices and we are displaying the monatomic lattice's band structure as folded in to the same Brillouin zone as that of the SSH lattice. Finally, the two lattice layers are uniformly coupled together, resulting in a system with four resonators per unit cell that full-wave simulations show to exhibit a gapless spectrum, see Fig. 1e. We refer to this two-layer lattice as an acoustic metallized SSH lattice. Note that the choice of lattice layer couplings (and identical resonator geometry) ensures that the full system respects chiral symmetry, which is necessary to allow for the possibility of topological states at mid-spectrum that are associated with this local symmetry classification. (As our 1D lattice has real coupling coefficients, it is in class BDI of the Altland–Zirnbauer symmetry classification, which possesses an integer invariant in 1D[32,49,50]).

Our phononic crystals consist of acoustic cavities coupled together via grooved channels in the system's base and also via direct bridges (see Fig. 1a, b). The geometry of the cavities is designed so that their fundamental axial pressure modes are well-separated in frequency from the rest of the resonant spectrum. The coupling strength between adjacent resonators is controlled through the width of the channels. The base of the system is laser-cut, while the dimmers of bridged resonating cavities are fabricated by 3D printing UV-curing resin. In particular, a bipartite single-layer metamaterial consisting of identical resonators and alternating coupling channel groove widths yields an accurate acoustic realization of the SSH lattice[48] (see Supplementary Note 1). By designing a metamaterial with a domain boundary between two topologically distinct SSH insulating phases (without the added monatomic layer), and trivial outer edges, both simulation and experimental observations confirm that the system possesses a single topological resonant mode localized at the domain boundary, whose frequency is in the middle of the system's bulk band gap (see Fig. 2). Moreover, this topological mode's mid-gap frequency serves as implicit confirmation that the experimental platform can accurately reproduce chiral symmetry. To facilitate identification of the effects of chiral symmetry in our data we show squared frequencies, as these are the eigenvalues of the acoustic wave equation that are equivalent to energy in the Schrödinger equation.

The phononic topological metal is experimentally realized by adding a layer of identical acoustic resonators, with uniform intra-layer coupling strength $t_M$, to the SSH layer and coupling the two layers together via uniform nearest-neighbor couplings $t_c$, see Fig. 3a,b,c. In doing so, we are preserving the domain-wall boundary in the SSH layer and its trivial edge terminations, but there is no alteration to the monatomic layer at the domain boundary. Even so, this still yields a domain wall in the combined system. Despite the full system's gapless spectrum, both full-wave numerical simulations (Fig. 3d) and

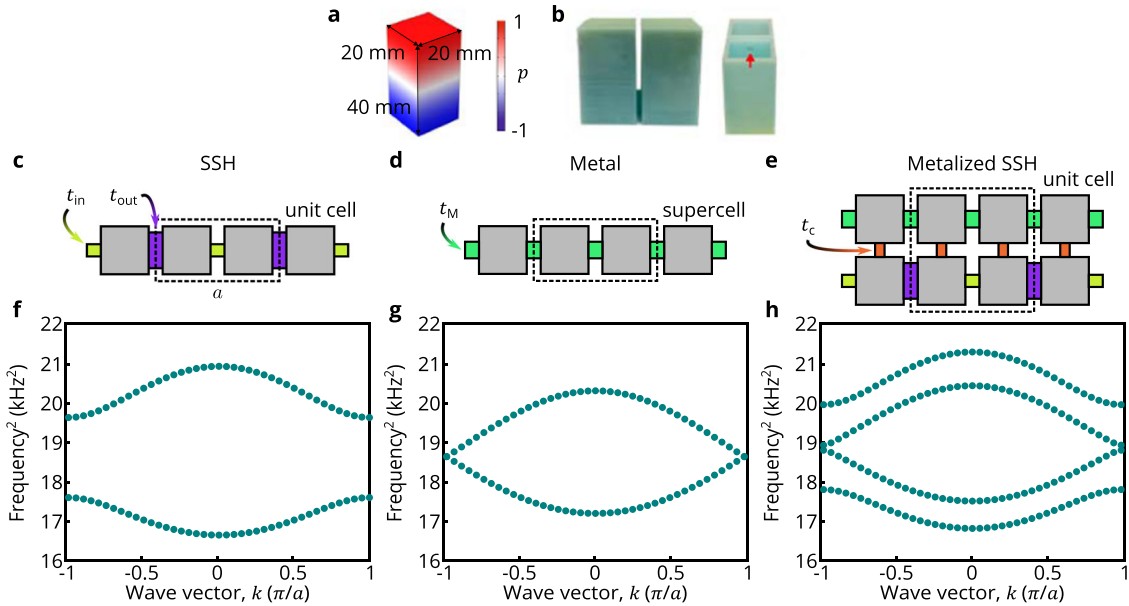

**Fig. 1 | Designing a phononic topological metal. a** Simulated acoustic eigen-pressure field $p$ for a single acoustic block resonator at the first elementary reso-nance mode with its geometry parameters, 20 mm by 20 mm by 40 mm. $p$ is shown in normalized units. **b** 3D-printed acoustic dimer consisting of one resonator belonging to the SSH layer and one resonator belonging to the metal layer, with the pair connected by an acoustic 3D printed bridge (red arrow points to one entrance to this bridge). The coupling bridge, parameterized by $t_c$, has width 3 mm, height 3 mm, and length 6 mm. **c, d, e, f, g, h** Schematics (**c, d, e**) and full wave simulations of the band structure (**f, g, h**) of the acoustic SSH lattice (**c, f**), acoustic metal lattice (**d, g**), and acoustic metallized SSH lattice (**e, h**). Band structures are shown in units of frequency squared to emphasize the symmetric spectrum due to chiral sym-metry. The wave vector $k$ lies in the first Brillouin Zone, which ranges from $-\pi/a$ to $\pi/a$, where $a = 52$mm is the lattice constant. The couplings in the SSH lattice $t_{in}$ and $t_{out}$ are defined by channels with widths 15 mm ($t_{in}$) and 5 mm ($t_{out}$), and the same height 3 mm, and length 6 mm. The metal layer's coupling $t_M$ stems from a channel with dimensions width 7 mm, height 3 mm, and length 6 mm.

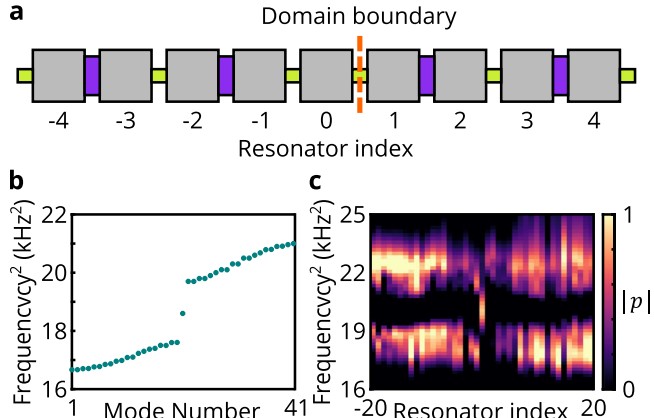

**Fig. 2 | Observation of a phononic SSH lattice with a domain wall. a** Schematic of the domain wall in the phononic SSH lattice. Both edges of the lattice have topo-logically trivial terminations. **b** Full-wave simulations of the system's eigen-frequencies with 41 lattice sites in total. **c** Measured local density of states (LDOS), assembled from microphone readings on 41 resonators of the acoustic SSH system. The observed pressure amplitude $|p|$ is shown in normalized units. A bulk band gap can be identified and the resonant mode in the bulk band gap is the domain boundary mode.

experimental observations (Fig. 3e,f) show that this metamaterial possesses edge- and domain wall-localized resonant states, whose appearance is linked to the relative strength of the coupling coeffi-cients. In particular, this system exhibits four boundary-localized states in total, two that are approximately localized at the system's domain wall, and one at each edge, see Fig. 3f, g. Moreover, we note that the phononic topological metal has many more topological states than one might initially expect—by itself, the SSH layer has topologi-cally trivial edges (there are no edge-termination-localized states in Fig. 2c) and a single domain-wall-localized state.

However, it is not possible to use topological band theories to predict the appearance of the four boundary-localized states seen in Fig. 3f,g. Attempts to define a winding number (or another similar integer invariant for 1D chiral-symmetric systems) based on this crys-tal's bulk structure cannot work; the lack of a bulk band gap would require a path of matrix determinants that tried to characterize this winding to intersect the origin, making the winding number undefined[51]. Moreover, the topology of the observed boundary-localized resonances in the finite system cannot be understood from the presence of low-dimensional degeneracies in the periodic gapless system's spectrum, as is the case for topological semimetals. For a $d$-dimensional semimetal, its topology is connected to features in its band structure with dimension $\leq d-2$, and results in states on its $(d-1)$-dimensional surfaces that can be identified within an incom-plete band gap. In contrast, our 1D periodic phononic topological metal (Fig. 1e) cannot possess such low-dimensional band features, and upon introducing an edge or domain wall completely lacks any such kind of incomplete band gap (Fig. 3c,d). Likewise, as the observed boundary-localized states are at mid-spectrum, we seek a topological classification that predicts and protects this spectral location, which precludes crystalline invariants[28].

Altogether, standard theories of topology are unable to distin-guish between the two bulk phases in this system and identify whether these localized states are of topological origin, or provide a measure of topological protection.

## Theory of the spectral localizer

Instead, to prove that the observed localized states are of topological origin and that their existence can be tied to a bulk-boundary corre-spondence, we use the spectral localizer as this approach can be applied to systems that lack a bulk band gap[24]. In general, a system's spectral localizer combines its Hamiltonian and position operators using a non-trivial Clifford representation. However, as our acoustic metamaterial is a 1D chiral symmetric system in which all of the

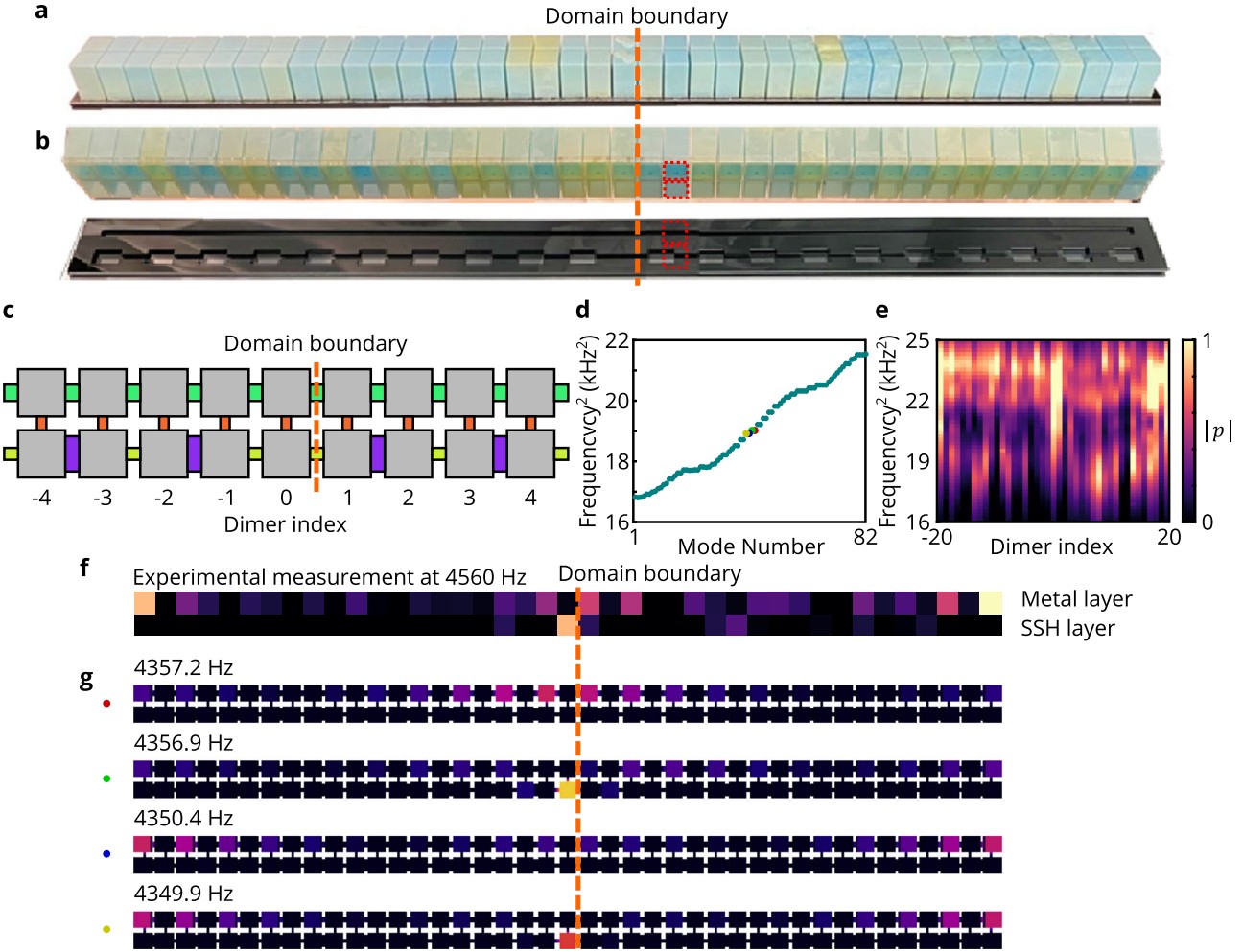

**Fig. 3 | Experimental demonstration of phononic topological metal.**
**a** Photograph of the fully assembled acoustic metallized SSH lattice consisting of block resonators and coupling bridges. **b** Photograph of the inner structure, with all coupling bridges visible. The 3D-printed resonators are open at the bottom and are coupled through the 3D-printed bridges and grooves in the black base when assembled. The acoustic resonators are embedded in the transparent acrylic plate (top) that sits above the black base (bottom). The red dotted squares indicate how the acoustic resonator dimers are mounted on the black base. **c** Schematic of the phononic metallized SSH lattice showing the domain wall. The SSH layer is terminated in the same manner as Fig. 2. **d** Full-wave eigenfrequencies of the metallized

SSH system consist of 41 dimers. **e** Measured local density of states, assembled from microphone readings on the 41 dimers of the metallized SSH system. The observed pressure amplitude |p| is shown in normalized units. LDOS is resolved by the dimer index. The data from the SSH resonator and the data from the metal resonator with the same dimer index are summed together. **f** Measured acoustic pressure field distribution at 4560 Hz for the domain boundary- and edge-localized modes. **g** Simulated acoustic eigenpressure fields of the four domain boundary- and edge-localized states. The colored dots in this panel correspond to those marked in **c**. The color indicates the absolute acoustic pressure |p|, which is displayed in normalized units.

couplings are real (i.e., its effective Hamiltonian is real-symmetric), its spectral localizer can be written in a reduced form as (see Supplementary Note 2)

$$\tilde{L}_{(x,E)}(X,H) = \kappa(X - xI)\Pi + H - iE\Pi. \tag{1}$$

Here, the spectral localizer can be evaluated at any choice of parameters $x, E \in \mathbb{R}$ (inside or outside of the system's spatial and spectral extent), $\kappa$ is a tuning parameter that also ensures that the terms have compatible units, $\Pi$ is the system's chiral operator, $H\Pi = -\Pi H$, and $I$ is the identity matrix. Although the spectral localizer is basis independent, if $H$ is written in a tight-binding basis, $X$ is simply a diagonal matrix whose entries correspond to the coordinates of each lattice site.

The spectral localizer (of appropriate dimension) can be used to both construct the relevant local topological invariant for a system in any symmetry class, as well as define the associated local gap. For the two-layer phononic topological metal considered here (or any other 1D

system in class BDI), its local invariant is given by[25],

$$\nu_L(x) = \frac{1}{2} \text{sig}\left(\tilde{L}_{(x,0)}(X,H)\right) \in \mathbb{Z}, \tag{2}$$

where sig is the signature of a matrix, i.e., its number of positive eigenvalues minus its number of negative ones. Note, $\nu_L(x)$ is only defined for $E = 0$, which reflects the fact that chiral symmetry can only protect states at the middle of the system's spectrum. Similarly, the local gap $\mu_{(x, E)}$ is given by the smallest singular value of $\tilde{L}_{(x,E)}$,

$$\mu_{(x,E)}(X,H) = \sigma_{\min}(\tilde{L}_{(x,E)}(X,H)). \tag{3}$$

More generally, $\mu_{(x, E)}$ is used to define the Clifford pseudospectrum of $(X, H)$[52].

Together, $\nu_L(x)$ and $\mu_{(x, 0)}$ yield a complete picture of a system's topology. Rigorously, $\nu_L(x)$ is ascertaining whether the matrices $H$ and $X - xI$ can be continued to the chosen trivial atomic limit while

preserving both operators' real-symmetric form and chiral symmetry, and without closing the associated local gap, i.e., $\mu_{(x,0)} > 0$ during the entire continuation process; if $v_L(x) = 0$, such a continuation is possible. (Here, the limit that is considered trivial is specified by the choice of $\Pi$ in Eq. (1)). Moreover, this picture of topology is entirely local, different choices of $x$ can yield different invariants—for $x$ sufficiently far outside of the system's spatial extent, one expects to see a system with trivial local topology, while $x$ in the bulk of the system may reveal non-trivial local topology. At the domain boundary between these two regions, $x_0$, where $v_L(x_0)$ changes, the local gap must close $\mu_{(x_0,0)} = 0$, which is a direct manifestation of bulk-boundary correspondence[33] and is an indication that there are eigenstates or resonances of $H$ near $x_0$ at $E = 0$[52].

When the Hamiltonian itself displays a global spectral gap, the topological invariant supplied by the spectral localizer coincides with the traditional 1D winding number[51] (or, more generally, even/odd Chern numbers or $\mathbb{Z}_2$ invariants, depending on the dimensionality and symmetry of the system[26,27]). In the absence of such a gap, $v_L(x)$ has no analogue in the traditional way of applying $K$-theory: The spectral localizer simply pushes the applicability of the $K$-theoretic methods to previously unclassifiable systems and, in the present context, provides the means to formulate a bulk-boundary correspondence principle involving interface resonances as opposed to infinitely lived bound states. By a mechanism somewhat similar to one in complex scaling[53], the spectral localizer pushes away the continuum spectrum of $H$ by opening a gap at locations away from the position being probed $x$, allowing for the study of spectral flows and their associated topology.

Applying the spectral localizer to a tight-binding approximation of the acoustic metallized SSH model proves that the localized states observed in this lattice are connected to a bulk topological invariant (see Supplementary Note 6). In particular, we numerically observe the local invariant $v_L(x)$ in this system to change a few times, both at the system's boundaries and twice at the domain wall within the lattice's interior. Moreover, despite the fact that this lattice does not possess a bulk band gap, for most values of $x$ we find that the localizer $\tilde{L}_{(x,E)}(X,H)$ does have a reasonable spectral gap at $E = 0$ that protects the bulk topological invariant.

## Using the localizer as a system Hamiltonian

Beyond numerically calculating the phononic topological metal's local invariant and associated strength of protection, the form of the spectral localizer at $E = 0$ also inspires an experimental approach to verify the system's topology directly. In particular, because the reduced spectral localizer Eq. (1) is Hermitian at $E = 0$, it can be reinterpreted as a set of Hamiltonians itself, with

$$\tilde{H}_x \equiv \tilde{L}_{(x,0)}(X,H) = \kappa(X - xI)\Pi + H, \tag{4}$$

in which $\kappa(X - xI)\Pi$ is now an on-site potential with a sign that is sublattice-dependent (i.e., a modification of the central frequencies of each resonator), and the choice of $x$ re-centers this potential at a different lattice site (or anywhere in between lattice sites). Thus, by simulating and observing the spectrum of $\tilde{H}_x$, we are directly measuring the spectrum of $\tilde{L}_{(x,0)}$, which, through Eqs. (2) and (3), determines the topology and associated protection of the underlying system described by $H$ at $x$.

In practice, this reinterpretation presents a challenge, as $\|X - xI\|$ can become arbitrarily large as the lattice's size increases, but it is not possible to alter a resonator's geometry to yield arbitrarily large or small resonance frequencies. Instead, we can circumvent this challenge by using the substitution

$$\kappa(X - xI)\Pi \rightarrow \kappa\left[\tanh\left(\frac{X - xI}{\alpha}\right)\right]\Pi, \tag{5}$$

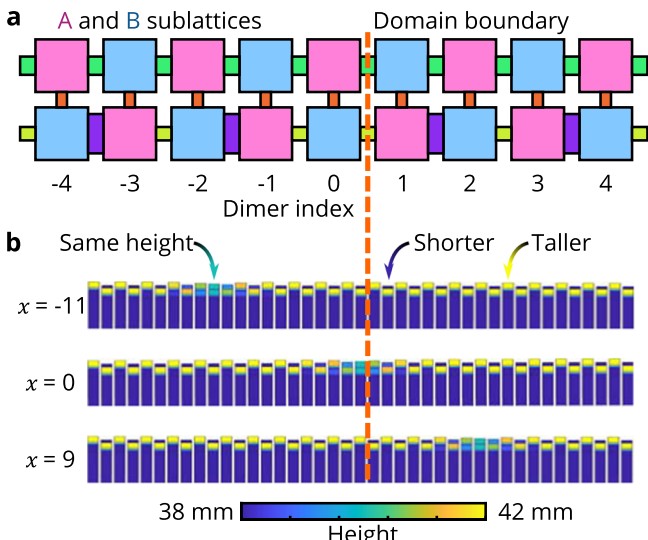

**a** A and B sublattices                                    Domain boundary

Dimer index
-4  -3  -2  -1  0  1  2  3  4

**b** Same height                                  Shorter   Taller

$x = -11$

$x = 0$

$x = 9$

38 mm                                              42 mm
Height

**Fig. 4 | Experimental protocol for observing the spectral localizer. a** Schematic of the spectrally localized phononic metamaterial with the domain wall shown. The two sublattices of the system are indicated in magenta and cyan, which correspond to entries of $+1$ and $-1$ in $\Pi$, respectively. **b** The configurations of the system when the localizer is centered at $x = -11$, $x = 0$, and $x = 9$. The height of each resonator above 38 mm is indicated by the color scale. As the underlying phononic topological metal in Fig. 3 uses resonators that are 40 mm tall, and small changes to the resonator volume change its frequency without changing its couplings, this coloration is effectively showing the on-site potential added in Eq. (6).

such that

$$\tilde{H}_x = \kappa\left[\tanh\left(\frac{X - xI}{\alpha}\right)\right]\Pi + H. \tag{6}$$

As this bounded operator is linear in the vicinity of $x = 0$ (the range of approximate linearity is set by $\alpha$), one can prove that it preserves the necessary information for determining the system's topology using Eq. (2) (see Supplementary Note 3). Moreover, this choice of alteration to the system's resonators can be experimentally realized for lattices of any size (see Fig. 4).

We directly confirm the topological behavior of the phononic topological metal described by the Hamiltonian $H$ by numerically and experimentally observing the properties of its spectrally localized counterpart given by $\tilde{H}_x$ at many different choices of $x$ (a realization for a single $x$ is shown in Fig. 5a). In particular, the sublattice-dependent on-site potential is realized by modifying all of the resonator heights by up to 2 mm, which preserves their well-separated fundamental axial mode but yields a shift their frequency (i.e., a different on-site potential). First, full-wave simulations of the localized metamaterial, $\tilde{H}_x$, demonstrate that there are four $x$ locations in the underlying system where the local invariant changes and the local gap closes (Fig. 5b,c,d), two locations at the outer edges of the system, and two next to the domain boundary. As locations where $\mu_{(x,0)} = 0$ predict the presence of states of $H$, the localized states seen in the original phononic topological metal (Fig. 3f,g) are a direct manifestation of bulk-boundary correspondence and are necessarily of topological origin. Thus, despite the absence of a bulk band gap in the system, these topological states are protected by the non-zero local gap $\mu_{(x,0)} \approx 0.1$ kHz$^2$ surrounding the locations where these states appear. In other words, chiral-preserving perturbations to the system $H \rightarrow H + \delta H$ cannot alter the local topology at $x$ so long as $\|\delta H\| < \mu_{(x,0)}$. Furthermore, we note that these states possess additional protection due to a relatively large secondary gap $\approx 0.4$ kHz$^2$, see and Fig. 5e and Supplementary Note 4. Heuristically, every gap in the localizer can be

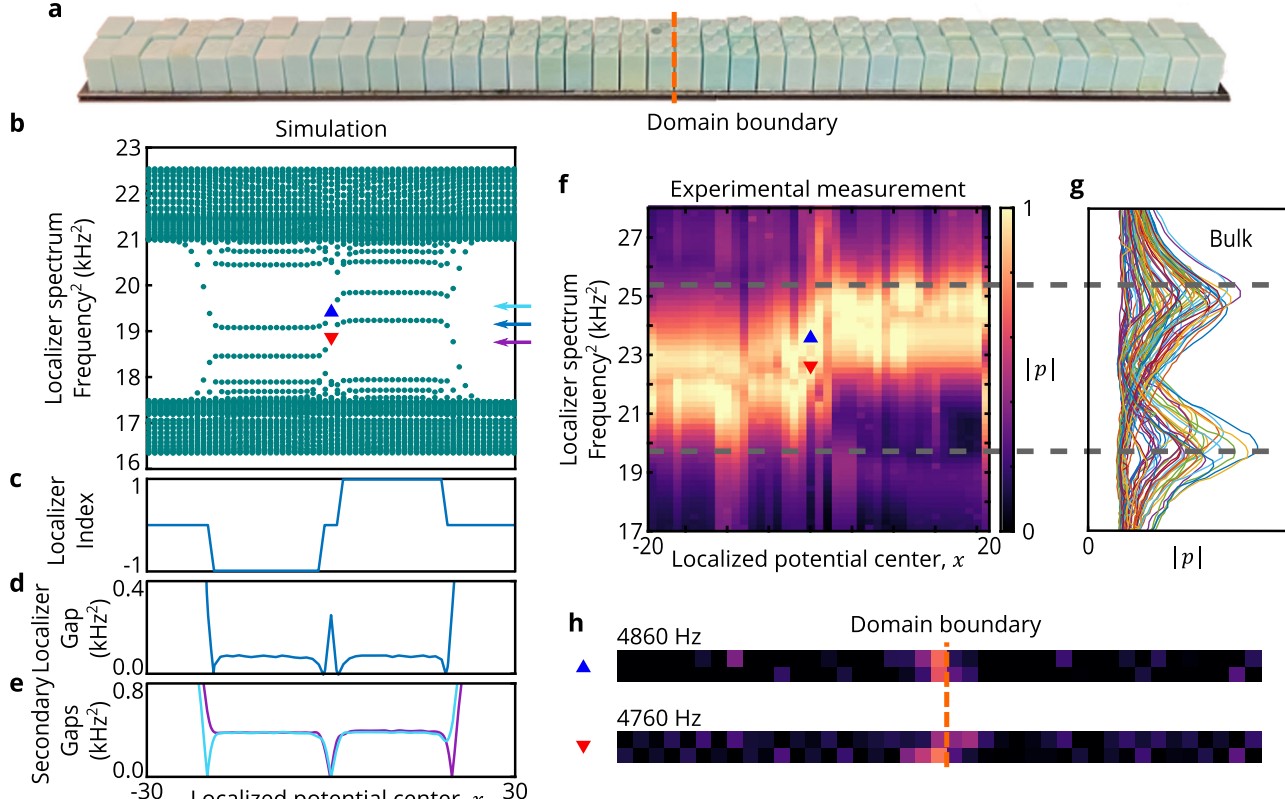

**Fig. 5 | Observation of topology using a spectrally localized acoustic meta-material. a** Photograph of the fully assembled spectrally localized metamaterial with the added sublattice-dependent on-site energies to the underlying metallized SSH lattice. The resonators that comprise this system can be re-assembled to realize different choices of the center of the localized potential $x$ in Eq. (6). **b** COMSOL simulated resonant spectrum of the spectrally localized metamaterial as the position of the localized potential's center $x$ is varied, demonstrating the existence of the two bands in the dynamical localization gap. **c,d,e** Localizer index (**c**), localizer gap (**d**), and secondary gaps (**e**) derived from the full-wave simulated spectrum. The localizer index and gap are calculated using the mid-spectrum frequency indicated in **b** (blue arrow on right), and the two frequencies chosen for calculating the secondary gaps are similarly indicated (cyan and magenta arrows on right). **f** Experimental mapping of the local density of states as the localized potential's center is moved (constructed from microphone readings on the dimer where the localized potential is centered $x$), confirming the existence of the two central eigenvalues for varying $x$ seen in **b**. $\alpha = 2.5a$ and $\kappa = 1.85$ kHz$^2$ in Eq. (6) were used in our simulations and experiments, where $a$ is the lattice constant. The observed pressure amplitude $|p|$ is shown in normalized units. **g** Measured pressure in normalized units for the spectrally localized system's bulk eigenvalues constructed from the microphone readings on the bulk resonators at least 3 resonators away from the localized potential center. The gray lines in **f** and **g** are showing that the spectrally localized system's two central sets of eigenvalues are well separated from the system's remaining eigenvalues. **h** Experimentally measured mode profiles at 4760 Hz and 4860 Hz when $x = 0$ using the same color map as **f**. The red and blue triangles in **b** and **f** correspond to the frequency and localized potential center chosen for observing these data.

associated with an element in a $K$-theory group, and thus can provide some form of topological protection[54]. Although most such gaps are too small for the resulting topology to be physically robust, for this particular system the secondary gap is relatively large and thus provides strong protection for one of the two states at the domain boundary at the system's center.

We experimentally realize the localized metamaterial, and directly characterize the changes in the underlying system's local $K$-theory (Fig. 5f, g, h). In particular, the simulated resonance spectrum is reproduced with high fidelity by experimental measurements, confirming the topological properties of our phononic topological metal. Likewise, the two central bands (Fig. 5f) are observed to be well separated from the remainder of the spectrally localized system's bulk bands (Fig. 5g). Although different choices of $x$ in $\tilde{H}_x$ yield distinct physical systems (see Fig. 4b), our acoustic metamaterial is re-configurable, and thus we do not need to fabricate a new system for each choice of $x$ shown in Fig. 5. The discrepancies between simulation and experiment are likely the result of fabrication imperfections and variations, as well as measurement errors. However, the discrepancy observed in Fig. 5 is still of a similar magnitude to those observed both Figs. 2 and 3. In particular, the observed discrepancy in the SSH lattice (Fig. 2) shows that these differences are standard to the acoustic

metamaterial platform, and are not substantially larger or smaller for our metallic (Fig. 3) or spectrally localized systems (Fig. 5).

Thus, altogether, our experimental results, coupled with our full-wave simulations, prove that our underlying phononic metamaterial (Fig. 3) is a gapless topological material.

## The underlying $K$-theory

The relative simplicity of the equations of the spectral localizer and the local topological invariants it provides tends to obscure the $K$-theory that it rests on. Thus, to show how our experimental protocol for altering a system to directly observe its topology stems from $K$-theory, we provide a brief discussion for an interested reader aimed at illuminating the spectral localizer's mathematical foundation. A traditional form of $K$-theory, topological $K$-theory[55,56], works with continuous functions that map from a given topological space to a space of structured matrices. These spaces of structured matrices are called classifying spaces, as they can be used to compute all 10 $K$-theory groups associated to the original given space. (Classifying spaces for classes of real or complex vector bundles would be special cases associated with this mapping.) A newer, more powerful form of $K$-theory is the $K$-theory of $C^*$-algebras[57], which still applies when one has momentum space but also applies when momentum space is lost.

Here, we work with modified forms of $C^*$-algebra $K$-theory[25,58–64] that are more directly applicable to finite systems. Moreover, these newer forms of $K$-theory lead to efficient numerical algorithms and are adaptable to different symmetry classes. The main speedup of these approaches comes from avoiding spectral flattening, as numerically, operations like projecting into an occupied subspace tend to produce dense matrices even when the original system can be described with sparse matrices.

Following Ref. 32, we briefly review how classical topological $K$-theory arises in the case of a periodic 1D system in class BDI. The position operator is used to define momentum space, which is a copy of the circle $\mathbb{T}^1$. The chirality of the Hamiltonian means

$$H(k) = \begin{bmatrix} 0 & U(k) \\ U^\dagger(k) & 0 \end{bmatrix}. \tag{7}$$

If we have taken the optional step of spectrally flattening $H$, we find that $U(k)$ is unitary. Thus, the topology in class BDI arises from attempting to classify the ways in which continuous functions can map from the circle to the classical groups of unitary matrices; that is, homotopy classes of elements of $C(\mathbb{T}^1, \mathcal{U}(n))$. Time-reversal symmetry manifests itself as $U^*(k) = U(-k)$. As Kitaev explains[32], homotopy classes in $C(\mathbb{T}^1, \mathcal{U}(n))$ can be used to form a group, one of the classic groups in topological $K$-theory[56]. Finally, the $K$-theory group element determined by $H(k)$ can be calculated using a winding number.

For non-periodic, finite 1D systems in class BDI, we use the form of $K$-theory that associates groups to certain algebras. The relevant algebra is $\mathcal{A} = \boldsymbol{M}_{2n}(\mathbb{C})$, treated as a graded, real $C^*$-algebra, and $2n$ is the number of sites in the system. The grading is determined by $\Pi$ and we use the standard reality structure (real matrix means real entries). The zeroth group of $K$-theory for this algebra $K_0(\mathcal{A})$ is built out of out of homotopy classes of $2n$-by-$2n$ unitary matrices $U$ that also satisfy $\Pi U \Pi = U^\dagger$ and $U^\top = U$[60]. However, if we want to avoid spectral flattening, we can instead look at homotopy classes of invertible matrices $M$ such that $\Pi M \Pi = M^\dagger$ and $M^\top = M$. Finally, the element of $K_0(\mathcal{A})$ determined by $M$ is calculated using half the signature of $M\Pi$.

To apply this general discussion to the spectral localizer, consider the full 1D spectral localizer at $E = 0$

$$L_{(x,0)}(X,H) = \begin{bmatrix} 0 & \kappa(X - xI) - iH \\ \kappa(X - xI) + iH & 0 \end{bmatrix}, \tag{8}$$

which at most positions $x$ will contain two invertible matrices, $M_x = \kappa(X - xI) + iH$ and its adjoint. Notice that $M_x\Pi$ is Hermitian (but not real), and thus we can determine the underling physical system's topology by calculating sig$(M_x\Pi)$. Moreover, it has been established[25] that in the BDI symmetry class $M_x\Pi$ is unitarily equivalent to the real symmetric matrix $\kappa(X - xI)\Pi + H$ (i.e., Eq. (4)), and thus sig$(M_x\Pi) = $ sig$\left(\tilde{H}_x\right)$, which is what is used in Eq. (2).

Altogether, a more standard approach to topological materials would consider the unitary $U_x$ that is derived from $M_x$ by spectral flattening (that is, $U_x$ is the unitary polar factor of $M_x$). Then one can apply the graded trace, which is more familiar in pure math than the graded signature. However, since

$$\mathrm{tr}(U_x\Pi) = \mathrm{sig}(M_x\Pi) = \mathrm{sig}\left(\tilde{H}_x\right) \tag{9}$$

we have many mathematically equivalent formulas to choose from to determine the underlying system's topology. In particular, the latter two of these formula can be immediately recognized from the invertible matrix form of $C^*$-algebra $K$-theory that underpin the spectral localizer. From the perspective of numerical efficiency, the formula involving $U_x$ would be the slowest, assuming one actually performs the spectral flattening. The formula involving $\tilde{H}_x$ will be the fastest, as this matrix will be real, symmetric, and (usually) sparse.

Thus, as $\tilde{H}_x$ is the Hamiltonian for the localized system (Fig. 5) that we realize to observe the topology of the underlying phononic topological metal (Fig. 3), our experimentally methodology is inextricably linked to the $K$-theory of $C^*$-algebras.

## Discussion

In conclusion, we have demonstrated a topological metal in an acoustic metamaterial and directly observed its boundary-localized states despite its lack of a bulk band gap. To do so, we have used the spectral localizer, a local theory of topological materials that is able to predict topological phenomena, and a measure of topological protection, even in the absence of a bulk band gap. Moreover, we have introduced an experimental protocol that uses a system's spectral localizer as its Hamiltonian, providing a direct probe of the underlying system's local $K$-theory. Here, it is worth emphasizing that this protocol can be applied to any topological system. Although our specific demonstration has leveraged the system's symmetries to yield a real-symmetric spectral localizer, the spectral localizer for any system is, by definition, Hermitian, and as such, it can always be adapted to be an observable system. Thus, the overall methodology that we have introduced may enable the prediction and observation of topological metals across a broad array of systems, including materials that exhibit higher-order topology and those whose topology is determined by its crystalline symmetries. Finally, as the spectral localizer takes an operator-based, rather than eigenstate-based, approach to topology, it is potentially broadly applicable to interacting systems, a class of systems that traditional band theories of topology have had difficulty gaining traction with.

## Methods

### Fabrication

Our fabrication process is modular and the acoustic crystals are assembled from parts that are independently manufactured with different automated process. This approach enables a high throughput of acoustic crystals, which can be disassembled and stored after use.

One leg of the process is the manufacturing of the supporting bases, which consist of two layers of 3-mm thick black acrylic plates (Fig. 3b) and one top layer of 1.5-mm thick transparent acrylic plates (Fig. 3a,b), of which the top transparent layer and middle black layer have through holes, laser-cut at specific geometries with the Boss Laser-1630 Laser Engraver. The middle layer with 3mm deep through channels provides coupling channels between the dimers. The top transparent layer with through square holes holds the dimers in place.

The resonators were manufactured using an Anycubic Photon 3D printer, which uses UV resin and has $47\mu m$ XY-resolution and $10\mu m$ Z-resolution. The thickness of their walls is 2 mm, to ensure a good quality factor and to justify rigid boundaries in our numerical simulations. The inner dimensions of the resonators are supplied in Fig. 1a,b. For the metallized-SSH system, the resonators are 3D printed as dimers with identical narrow channels connecting the resonators. To implement the spectral localizer, the resonators were printed with different heights according to the algorithm Eq. (6) and were made ready for the assembling.

The resonators were mounted and coupled through the channels grooved in the acrylic plates of the base. Let us specify that the resonators are interchangeable so that they can be move around and acoustic crystals with different probe positions can be generated, as described in the main text. Finally, we note that although the coloration of the resonators is not uniform in either of the structures shown in Fig. 3a,b or 5a, this is simply an artifact of the fabrication process, and these color differences do not impact their behavior in any way.

### Experimental protocols

The protocol for the acoustic measurements reported in Fig. 3 and Fig. 5 was as follows: Sinusoidal signals of duration 1 s and amplitude of

0.5 V were produced with a Rigol DG 1022 function generator and applied on a speaker placed in a porthole opened in a resonator. A dbx RTA-M Reference Microphone with a Phantom Power was inserted in a porthole opened in the same resonator where the speaker was inserted and acquired the acoustic signals. The signals were read by a custom LabVIEW code via National Instruments USB-6122 data acquisition box and the data was stored on a computer for graphic renderings.

The local densities of states reported in Fig. 3d and Fig. 5f,g were obtained by integrating the local density of states acquired from resonators whose index are the same as the position of the probe. Same instrumentation was used. The measurements were repeated with moving the position of the probe. For each measurement, the frequency was scanned from 4200 Hz to 5200 Hz in 10 Hz steps.

## Simulation

The simulations reported in Fig. 1, 2, 3, and 5 were performed with the COMSOL Multiphysics pressure acoustic module. The wave propagation domains shown in Fig. 1 were filled with air with a mass density 1.3 kg/m$^3$ and the sound's speed was fixed at 343 m/s, appropriate for room temperature. Because of the huge acoustic impedance mismatch compared with air, the 3D printing UV resin material was considered as hard boundary.

## Data availability

All the data generated in this study have been deposited in the Zenodo database https://zenodo.org/record/7765335#.ZB3G-nbMLD6.

## Code availability

The computer codes used in this study have been deposited in the Zenodo database https://zenodo.org/record/7765335#.ZB3G-nbMLD6.

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

## Acknowledgements

T.L. acknowledges support from the National Science Foundation, grant DMS-2110398. This work was performed, in part, at the Center for Integrated Nanotechnologies, an Office of Science User Facility operated for the U.S. Department of Energy (DOE) Office of Science. A.C. and T.L. acknowledge support from the Laboratory Directed Research and Development program at Sandia National Laboratories. Sandia National Laboratories is a multimission laboratory managed and operated by National Technology & Engineering Solutions of Sandia, LLC, a wholly owned subsidiary of Honeywell International, Inc., for the U.S. DOE's National Nuclear Security Administration under contract DE-NA-0003525. The views expressed in the article do not necessarily represent the views of the U.S. DOE or the United States Government. C. P. acknowledges support from the National Science Foundation, grant CMMI-2131759. E.P. was supported by the U.S. National Science Foundation through the grants DMR-1823800 and CMMI-2131760.

## Author contributions

W.C., assisted by S.-Y.C., performed the full-wave simulations, fabrication, and experimental characterization. A.C. designed the underlying physical model. A.C., E.P., and T.A.L. developed the mathematical framework for analyzing this model. All authors analyzed and interpreted the data. W.C. and A.C. wrote the manuscript with input from all authors. E.P., T.A.L., and C.P. supervised the project.

## Competing interests

The authors declare no competing interests
