## [Peer Review File · Nature Communications]

REVIEWER COMMENTS

Reviewer #1 (Remarks to the Author):

I thank the authors for their response to my questions and apologize for the tardiness of my report.

I think the manuscript has improved. But I am of two minds whether to recommend publication. On one hand it is nice that they employ their spectral localizer technique. Moreover, they use these results in a, albeit artificial, setup. On the other hand one may question whether observing this bound state in [indeed] a metallic phase in a highly controllable system is of the importance to warrant publication in Nature communications.

Although most issues have been resolved I am still not entirely convinced on the prominent role of K-theory in the paper. Indeed in the end the paper is about a bound state that is diagnosed with the spectral localizer technique and the role of K-theory is not that on the foreground as suggested.

Moreover, in the blue text the authors argue that the spectral localizer ascertains which state is trivial if there a multiple wannier vacua [read as in the ssh model]. I can see this because one fixes the boundary conditions [in fact choosing the origin]. But this choice is relative in the sense that it can be altered. The winding of one phase becomes winding in the other. The point is that the BBC is thus relative [to taking an origin] and hence statements as “THE trivial limit” see also line 247 are in some sense obscure.

Finally, although I think that the concept of spectral localizers is insightful the idea that the commutator of H and X_j shows the obstruction still seems to also be tractable to irreps. Indeed if I know the complete irrep configuration I think one can also derive the obstruction. Indeed, the gapless phases can be seen in the phase diagram as the borders between changing the irreps, see also sec IV of added ref 28. All in all I fully understand the case of the authors that they employ their nice technique but it does put in perspective if this route is essential and not another manner to evaluate a bound state in a metallic system. This is important as then the case for recommending would be more on the experimental implementation of this metallic phase, which is slightly less influential as the current draft suggest. However, I think it would be fair to have the authors comment on that.

Reviewer #2 (Remarks to the Author):

This paper partly demonstrates the theoretical model in Ref. 1, where the 2D Haldane lattice is coupled to a single-band triangular lattice. In this paper, the considered system simply couples to a one-dimensional SSH chain with another resonator chain. This manuscript claims the observation of topological metal in an acoustic metamaterial. This work is more advanced in the perspective of mathematical skills, but does not reveal novel physical properties. Therefore, this reviewer finds that this manuscript is not suitable for the high quality Nature Communications publication.

The reviewer would like ask more questions:

How do the authors define a topological metal here? The definition of topological metal and semi-metal is not clear in the manuscript. There is only one sentence about the minimal dimension of Fermi surface, which is not clear and intuitive. As for the extended SSH system in the manuscript, the characteristics of topological metal are not clearly expressed. The minimal dimension of Fermi surface of the system is not mentioned to further verify the topological metal. In addition, the authors claim that develop a new experimental modality, but this important message is not very clear in the manuscript. How can the experimentally measured domain boundary localized modes be verified to be topological, and the system can be characterized as topological metal? For experimental data, the frequency ranges in Figure 2 and Figure 4 are different. Are their system parameters consistent? What are the experimental setup differences between Figure 2 and Figure 4? The frequency ranges of simulation and experimental results in Figure 4 do not match.

Reviewer #3 (Remarks to the Author):

The work presents topological interface mode in a gapless acoustic crystal and interprets this phenomenon by using the concept of spectral localizer Inspired by K theory. A 2-layer SSH model is constructed as a demonstration of the physical implementation of the proposed experimental protocols. The paper is well-organized with nice writing. The theoretical part is solid and may provide a new avenue to explore topological metals, while the demonstration of the acoustic “metalized SSH” model is not persuasive. I am afraid I cannot recommend it for publication in its current form.

- The definition of “metal layer” is not convincing. The layer can be regarded as a special case of the SSH model or a conventional acoustic crystal, which is not necessarily a metal phase.
- If the unit size of the “metal layer” change a bit, which is larger or small than the one in “The SSH layer”, the band structure with degeneracy at the end of wavevectors remains, but with some frequency shift. Thus, the interface mode can be well-isolated from the bulk modes in the “metalized SSH”

systems. I doubt that this system still can be treated as a “metalized SSH” model and if it is necessary to explain it by the theory of “spectral localizer”.

- It would be good if the coordinate axis can be added to Figures 1a and 1b, providing a better illustration of the design. Some of the resonator here tends to be yellow and some of them blue, what is the color indicates? Besides, where is the white dotted line mentioned in the caption? Some labels in Figure 1c are deformed. In addition, the dimer index is a bit confusing. If each dimer is counted as two, why the index starts from ± 1 instead ± 2 ? What does “n” indicate in Figure 1d? The position of the marker “2mm” is inaccurate and the value is inconsistent with the one mentioned in the caption.
- For the dispersion, why “frequency²” is used instead the “frequency” with physical meaning?
- For Figure 2b, as introduced by the authors, the SSH model consists of 41 resonators with the metal layer and the connection removed. Why the resonator index still spans from -41 to 39?
- It will be good to provide the full term of “BDI” on Line 189.
- The labels of Equation 4 and the following equations are missing.
- Please check the typos, e.g., “localizer’r”, “transparentr”.

Reviewer 1:

I thank the authors for their response to my questions and apologize for the tardiness of my report.

I think the manuscript has improved. But I am of two minds whether to recommend publication. On one hand it is nice that they employ their spectral localizer technique. Moreover, they use these results in a, albeit artificial, setup. On the other hand one may question whether observing this bound state in indeed a metallic phase in a highly controllable system is of the importance to warrant publication in Nature communications.

We begin by thanking the reviewer for their time and constructive feedback that enabled us to significantly improve the manuscript.

Although most issues have been resolved I am still not entirely convinced on the prominent role of K -theory in the paper. Indeed in the end the paper is about a bound state that is diagnosed with the spectral localizer technique and the role of K -theory is not that on the foreground as suggested.

We apologize for going too far in hiding the modern K -theory used. One of the advantages of the spectral localizer is that its indices can be understood without knowledge of vector bundles or C^* -algebras, but we appreciate the reviewer's point that some readers will want to see the connection to K -theory explicitly.

To address this, we have added a new subsection to clarify that we are using the K -theory of graded, real C^* -algebras. We have also added references to a few papers that discuss how the spectral localizer leads to K -theory invariants that are equivalent to more established invariants, at least in the limited cases where both the previously known invariants and the new localizer-based invariants can be applied. Finally, we would argue that given Eq. (4)'s origins in the form of modern K -theory discussed in this new subsection, the new experimental protocol that we have developed is a direct reflection of the prominence of K -theory in our work.

Moreover, in the blue text the authors argue that the spectral localizer ascertains which state is trivial if there a multiple wannier vacua [read as in the ssh model]. I can see this because one fixes the boundary conditions [in fact choosing the origin]. But this choice is relative in the sense that it can be altered. The winding of one phase becomes winding in the other. The point is that the BBC is thus relative [to taking an origin] and hence statements as "THE trivial limit" see also line 247 are in some sense obscure.

Of course. We have clarified the text to discuss a "chosen trivial atomic limit" for these classes of topology, rather than any sort of absolute trivial limit.

To address this comment, we have revised the text in the introduction (2nd to last paragraph), and section II.B (2nd to last paragraph of that subsection).

Finally, although I think that the concept of spectral localizers is insightful the idea that the commutator of H and X_j shows the obstruction still seems to also be tractable to irreps. Indeed if I know the complete irrep configuration I think one can also derive the obstruction. Indeed, the gapless phases can be seen in the phase diagram as the borders between changing the irreps, see also sec IV of added ref 28. All in all I fully understand the case of the authors that they employ their nice technique but it does put in perspective if this route is essential and not another manner to evaluate a bound state in a metallic system. This is important as then the case for recommending would be more on the experimental implementation of this metallic phase, which is slightly less influential as the current draft suggest. However, I think it would be fair to have the authors comment on that.

Here, we disagree with the reviewer — we do not think it will generally be possible to use the techniques outlined in PRX 7, 041069 (2017) (Ref. 28 in the manuscript) to classify the topology that protects the mid-spectrum, boundary-localized states that we observe in our phononic topological metal. Moreover, any invariants that arose from such a crystalline classification of our system would provide a different kind of

topological protection than what we are discussing.

Broadly speaking, this previous PRX study shows how to classify the crystalline topology of materials using arguments based in representation theory. The wavefunction spatial profiles of each band at the crystal's high symmetry points (HSPs), together with the available point group symmetries at each HSP and connectivity conditions for wavefunctions between the HSPs, define a set of integers (and relations between these integers) for a given material. These integers (and associated relations) can then be used to form a set of invariants that fully classifies the material's crystalline topology. However, this crystalline classification is *in addition to* the possible classification due to the system's local symmetry (i.e., the well-established ten-fold way classes, see Refs. 32, 49, and 50 in the main text), and is *not* a replacement of the existing ten-fold classifications. Indeed, the authors of the PRX write this explicitly for 2D class A systems used as the primary example in their study, "... [this] combinatorial argument does not indicate the possible values of Chern numbers, which need to be included in a full classification of topologically distinct phases." The two classifications are also independently listed in Table 1 of that work, and must be considered together to find the complete classification.

Thus, as our work is focused entirely on the topological classification of our system due to its chiral symmetry (i.e., one of the ten-fold way classifications), it exists outside of the framework of the kinds of crystalline topological invariants predicted by PRX 7, 041069 (2017). In other words, no amount of knowledge of the phononic topological metal's irreducible representations (irreps) will be able to show that its boundary-localized states are protected by chiral symmetry and are guaranteed to exist at mid-spectrum. (Just as arguments based on chiral symmetry cannot generally predict crystalline topology.) Moreover, the topological protection that we predict for our system is protection against chiral-preserving disorder; topological protection provided by a crystalline invariant protects against crystalline-symmetry-preserving disorder.

Finally, with respect to the gapless phases mentioned by the reviewer in Sec. IV of the PRX, to the best of our understanding these arguments are predicting protected semi-metal phases, not metal phases. As such, they would not be applicable to our system.

To address this comment we have added some text to the last paragraph of section II.A.

Reviewer 2:

This paper partly demonstrates the theoretical model in Ref. 1, where the 2D Haldane lattice is coupled to a single-band triangular lattice. In this paper, the considered system simply couples to a one-dimensional SSH chain with another resonator chain. This manuscript claims the observation of topological metal in an acoustic metamaterial. This work is more advanced in the perspective of mathematical skills, but does not reveal novel physical properties. Therefore, this reviewer finds that this manuscript is not suitable for the high quality Nature Communications publication.

We begin by thanking the reviewer for their time and constructive feedback that enabled us to significantly improve the manuscript.

The reviewer would like ask more questions:

How do the authors define a topological metal here? The definition of topological metal and semi-metal is not clear in the manuscript. There is only one sentence about the minimal dimension of Fermi surface, which is not clear and intuitive.

As discussed in the manuscript's introduction, we define 'topological metal' to specifically refer to systems that exhibit boundary-localized states whose origin can be predicted using a topological invariant calculated in the system's bulk (i.e., these materials possess bulk-boundary correspondence), but the boundary-localized states or resonances are degenerate in both energy and wave vector with bulk states.

Heuristically, a topological insulator's topological states exist in a complete band gap, and can be unambiguously identified at any wave vector (i.e., any remaining "good" wave vector once the material is truncated to contain an edge). A topological semi-metal's topological states exist in an incomplete gap, but can still be unambiguously identified at some wave vector(s). For example, the Fermi arc states of a Weyl semi-metal exist in an incomplete band gap, and they can be identified at energies and wave vectors (E, \mathbf{k}) within the incomplete band gaps in which they reside.

In contrast, under our definition of a topological metal, the topological boundary-localized state or resonance is degenerate with bulk states at all (E, \mathbf{k}) , for any possible truncation. Our particular (infinitely periodic) system is 1D, and possesses a gapless spectrum, see Fig. 1e. Thus, when the system is truncated (or a domain wall is inserted), there are no good remaining wave vectors, see Fig. 3d. As the topological state we observe has an energy within this gapless bulk spectrum, our system is a topological metal — the appearance of these states is related to a shift in a topological invariant calculated in the system's bulk.

To address this comment, we have revised some of the language in the manuscript's introduction, and rewritten parts of section II.A and II.C.

As for the extended SSH system in the manuscript, the characteristics of topological metal are not clearly expressed. The minimal dimension of Fermi surface of the system is not mentioned to further verify the topological metal.

Our original goal with the discussion of the 'minimal dimension of the Fermi surface' was to easily distinguish our system from topological semi-metals. However, we agree that this concept was not sufficiently developed in the previous draft of the manuscript.

To avoid confusion, and to make our manuscript accessible to a broad audience, we have removed this technical language. Instead, we have provided a much more detailed explanation of this set of concepts at the end of section II.A, where we discuss how it is not possible to understand our system's topology using any previous theoretical framework.

In addition, the authors claim that develop a new experimental modality, but this important message is not very clear in the manuscript. How can the experimentally measured domain boundary localized modes be verified to be topological, and the system can be characterized as topological metal?

Overall, the mathematical observation that underpins this new experimental modality is that the reduced spectral localizer that can determine our system's topology, Eq. (1) in the manuscript, is Hermitian when $E = 0$. Thus, this operator is not just an artificial construct, but can be treated a real system Hamiltonian in its own right, as we do in Eq. (4), and Figs. 4 and 5. In particular, all that must be changed for the system is to add on-site energies whose sign depends on whether the site is in the A or B sublattice of our chiral symmetric system.

The point is, that by tweaking our original system by just a little bit (i.e., making some of the resonators slightly smaller and others slightly larger, but keeping all of the connections the same), we can observe the spectrum of this altered system to directly see the topology of the original system (where all the resonators have the same geometry). As such, we can experimentally verify the original system to be a phononic topological metal.

Moreover, going forward, one could imagine performing this type of perturbation in other classes of systems with a spatially dependent potential, such as an applied electric field gradient in natural materials, or using a spatially-dependent optical pump to change the local index of refraction in a topological photonic system.

To address this comment, we have rewritten large portions of section II.C to make this new experimental modality more intuitive and transparent.

For experimental data, the frequency ranges in Figure 2 and Figure 4 are different. Are their system parameters consistent? What are the experimental setup differences between Figure 2 and Figure 4? The frequency ranges of simulation and experimental results in Figure 4 do not match.

As discussed in the previous response point, these two systems are not the same. The original phononic topological metal in Fig. 3 (in the current manuscript version, was previously Fig. 2) is comprised of resonators with identical geometry, while the spectrally localized phononic metamaterial in Fig. 5 (previously Fig. 4) consists of resonators that alternate between being larger or smaller.

Due to several factors, the frequency ranges of the simulation and experimental results do not always match. One of the main reasons for this difference is experimental error, which arises from the manufacturing and testing process. Manufacturing errors include imperfections in materials or components, variations in the manufacturing process, and limitations in the accuracy of measurement tools. Test errors are caused by fluctuations in environmental conditions, inaccuracies in measuring equipment. Despite these differences, the agreement between experiment and simulation in Fig. 5 is better than 85 %. Moreover, we would note that similar differences in the simulated and experimental data are seen in both the standard (unaltered) SSH lattice, shown in Fig. 2, as well as our phononic topological metal, Fig. 3, and with the added localized potential, Fig. 5. In particular, the observed discrepancy in the SSH lattice shows that these differences are standard to this acoustic metamaterial platform, and are not substantially larger or smaller for our metallic system.

To address this comment, we have rearranged and relocated many of the constituent subfigures within all of our figures. We believe these changes have made the figures easier to parse, especially by placing the photographs of the experimental systems right next to their associated LDOS measurements. We have also added a comment about these discrepancies to the main text near the end of section II.C.

Reviewer 3:

The work presents topological interface mode in a gapless acoustic crystal and interprets this phenomenon by using the concept of spectral localizer Inspired by K theory. A 2-layer SSH model is constructed as a demonstration of the physical implementation of the proposed experimental protocols. The paper is well-organized with nice writing. The theoretical part is solid and may provide a new avenue to explore topological metals, while the demonstration of the acoustic “metalized SSH” model is not persuasive. I am afraid I cannot recommend it for publication in its current form.

We begin by thanking the reviewer for their time and constructive feedback that enabled us to significantly improve the manuscript.

The definition of “metal layer” is not convincing. The layer can be regarded as a special case of the SSH model or a conventional acoustic crystal, which is not necessarily a metal phase.

Topological band theory and the spectral localizer are theories that describe properties in non-interacting systems. Thus, what we mean by “metal” is that the d -dimensional system has a gapless band structure and, specifically, has a Fermi surface that is always $(d - 1)$ -dimensional for all energies within its spectral extent. In contrast, a d -dimensional semi-metal has a bulk Fermi surface that is $\leq (d - 2)$ -dimensional, while an insulator does not have a Fermi surface. For our acoustic metamaterial, the Fermi surface is understood to be the energy of the topological state. A further discussion of this is in Chiu et al., Rev. Mod. Phys. 88 035005 (2016) (which is Ref. 5 in the manuscript), see section V, and in particular section V.A and Fig. 11 therein.

The concept we are distinguishing is that for a topological state in an insulator, there are no available bulk modes that the state could hybridize with. For a topological boundary-localized state in a semi-metal, there will be some wave vectors for which the topological state exists that will also be without available bulk states (so no hybridization will be possible for these wave vectors). For a specific semi-metal example, Fermi arc surface states in Weyl semi-metals exist in incomplete band gaps, they can be uniquely identified at some set of energy and wave vector pairs (E, \mathbf{k}) . In contrast, for a metal, the topological state will always have available bulk states it can hybridize with, for any choice of wave vector (i.e., the topological state will always be a member of a degenerate subspace at any pair (E, \mathbf{k})).

Thus, our “metal layer” meets the definition for being a non-interacting metal — it is a 1D system and for every energy within its band, there will be a 0D Fermi surface. Moreover, when the system is truncated or a domain wall is introduced, which breaks the system’s periodicity, there is no spectral gap for any wave vector. (There was only 1 periodic direction to begin with, so once that periodicity is broken by the domain wall / boundary there are no remaining “good” wave vectors.) As this gapless-ness is preserved when the metal layer is added to the SSH layer in our metamaterial, any topological state within the energy range of the gapless band structure will have available bulk states to hybridize with.

To address this comment, we have revised the manuscript’s introduction and provided a more thorough discussion of “topological metal” and “metal layer” in section II.A.

If the unit size of the “metal layer” change a bit, which is larger or small than the one in “The SSH layer”, the band structure with degeneracy at the end of wavevectors remains, but with some frequency shift. Thus, the interface mode can be well-isolated from the bulk modes in the “metalized SSH” systems. I doubt that this system still can be treated as a “metalized SSH” model and if it is necessary to explain it by the theory of “spectral localizer”.

There seem to be two reasonable ways of interpreting this comment – “unit size” could be understood as resonator geometry, and it could be understood as lattice constant (i.e., unit cell size). We will answer both cases. Overall though, both of these options will generally result in breaking the system’s chiral symmetry, and thus removing the possibility of finding boundary-localized states whose appearance is connected to changes in the local winding number defined in Eq. (2), as the system’s Hamiltonian will no longer satisfy

$$HII = -III.$$

If changing the “unit size” is referring to increasing or decreasing the metal layer’s resonator volume to still be uniform within that layer, but different from the geometry in the SSH layer, then yes, this will shift the 2 bands of the metal layer to be centered at a different energy. With a large enough on-site energy change, the two bands from the metal layer can leave the spectral extent of the two bands from the SSH layer, and thus the SSH layer’s mid-gap mode can be unambiguously identified in the system’s local density of states. However, changing even one of the resonators’ geometry amounts to adding an on-site energy to that site, which breaks the system’s chiral symmetry. Thus, this resulting system will not exhibit topological modes whose protection is linked to a winding number (except in the limit where the two layers are effectively decoupled).

If changing the “unit size” is referring to altering the lattice constant of the metal layer relative to that of the SSH layer (i.e., such that there are 2.3 or 3 resonators in the metal layer for every 2 in the SSH layer), the key question becomes understanding what couplings between the resonators are changed as a part of this lattice constant shift. In particular, are additional couplings added between the two layers? Overall, chiral symmetry enforces that the system can be equally divided into two sublattices, and that *all* of the couplings in the system link two sites from different sublattices. In other words, if the sublattices are labelled A and B , all of the couplings t_i connect one A site and one B site – there are no A – A or B – B couplings. Thus, the requirement to preserve chiral symmetry to protect the topological modes significantly limits the possibility of changing one of the layer’s lattice constant relative to the other — to have chiral symmetry (and thus, the mid-spectrum topological modes that we observe), the two layers will have to have commensurate lattice constants and the couplings will have to be carefully chosen.

Altogether, we would not expect either of these routes of changing the metal layer’s unit size to generally yield a system with topological modes that are protected by chiral symmetry to be at mid-spectrum. Moreover, given that either of these possible changes would destroy the system’s chiral symmetry, the essential symmetry in the system for our study, we would argue that these kinds of alterations would mean the system is no longer a metallized SSH model of any variety.

To address this comment, we have clarified the restrictions that preserving chiral symmetry places on our design in the first paragraph of section II.A.

It would be good if the coordinate axis can be added to Figures 1a and 1b, providing a better illustration of the design. Some of the resonator here tends to be yellow and some of them blue, what is the color indicates? Besides, where is the white dotted line mentioned in the caption? Some labels in Figure 1c are deformed. In addition, the dimer index is a bit confusing. If each dimer is counted as two, why the index starts from ± 1 instead ± 2 ? What does “ n ” indicate in Figure 1d? The position of the marker “2mm” is inaccurate and the value is inconsistent with the one mentioned in the caption.

We have significantly revised all of the figures to address these issues. The coloration of the resonators is simply an artifact of their fabrication process and does not alter their performance — a note about this has been added to the fabrication subsection of the Methods section. Coordinate axes for the dimer index have been added to Figs. 2, 3, and 4.

For the dispersion, why “*frequency*²” is used instead the “frequency” with physical meaning?

This choice has been made to help reveal the system’s topological properties. As the wave equation for acoustic systems is a Helmholtz equation rather than a Schrodinger-type equation, the eigenvalues of a phononic metamaterial’s effective Hamiltonian satisfy $H|\psi_n\rangle = \omega_n^2|\psi_n\rangle$, as opposed to $H|\psi_n\rangle = E_n|\psi_n\rangle$ for a natural material. Thus, we are choosing to display ω_n^2 as this preserves expected spectral features associated with H , whereas displaying ω_n would distort these features. In particular, as our system is chiral symmetric, one expects the spectrum of ω_n^2 to be symmetric about its center, but this will not be the case for ω_n . To address this comment, we have added a statement to the end of the 2nd paragraph of section II.A.

For Figure 2b, as introduced by the authors, the SSH model consists of 41 resonators with the metal layer and the connection removed. Why the resonator index still spans from -41 to 39?

This has been corrected.

It will be good to provide the full term of “BDI” on Line 189.

As the reviewer is undoubtedly aware, BDI is not an acronym, but instead is the designation of one of the ten matrix classes identified by Altland and Zirnbauer (see Refs. 32, 49, and/or 50). They adopted the rather random notation used by Élie Cartan and mathematicians in classifying simple graded Lie algebras, BDI, EI, EII, AIII, etc. To address this comment, we have added the full set of local symmetries and properties that matrices in class BDI possess to ensure clarity for a general audience, see the last two sentences of the first paragraph of section II.A.

The labels of Equation 4 and the following equations are missing.

These labels have been added.

Please check the typos, e.g., “localizer’r”, “transparentr”.

We appreciate the reviewer pointing these typos out. We have corrected these, and done our best to ensure that no other typos have crept in to the current manuscript.

REVIEWERS' COMMENTS

Reviewer #1 (Remarks to the Author):

I thank the authors for their insights. I must say I find the quality of the response high.. I agree on the K-theory answer; I was more trying to point out that in the crystalline context also gapless states can be addressed as stated, but am aware that the authors consider the chiral ten fold way class.

The theory is well fleshed out and matches the expertise of the authors. For publication I am still of two minds. The theory itself does not match this journal but of course is backed by an experimental part. The fact that the experiments are reported makes the paper more suited. However the SSH model and 'topological metal' phase still raises concern, as brought up before and also by the other referees, given the many works on this simple model [albeit that the authors slightly extend]. As a result I could see the sum of parts could be agreeable for publication although it is also true that the strongest part of the paper, being the theory, in itself would not be suited [also published in the related works] and that the extended SSH model is not very convincing in terms of novelty.

Reviewer #2 (Remarks to the Author):

The author has already answered the question, resolved my doubts, and I agree to publish this article.

Reviewer #3 (Remarks to the Author):

The changes the authors made have improved the manuscript and the response has addressed my previous concerns. I would like to provide my recommendation for its publication.

Dear Dr. Jakub Jadwiszczak,

We greatly appreciate the reviewers' thorough reports and thank them for their time spent helping us improve our study. Below, we have reproduced their brief reports for the revised manuscript. Additionally, all edits to the manuscript to make those changes asked for in the editorial requests document are marked in blue in the revised version.

Thank you,

Wenting Cheng, Alexander Cerjan, Ssu-Ying Chen, Emil Prodan, Terry A. Loring, and Camelia Prodan

Reviewer 1:

I thank the authors for their insights. I must say I find the quality of the response high.. I agree on the K-theory answer; I was more trying to point out that in the crystalline context also gapless states can be addressed as stated, but am aware that the authors consider the chiral ten fold way class.

The theory is well fleshed out and matches the expertise of the authors. For publication I am still of two minds. The theory itself does not match this journal but of course is backed by an experimental part. The fact that the experiments are reported makes the paper more suited. However the SSH model and 'topological metal' phase still raises concern, as brought up before and also by the other referees, given the many works on this simple model [albeit that the authors slightly extend]. As a result I could see the sum of parts could be agreeable for publication although it is also true that the strongest part of the paper, being the theory, in itself would not be suited [also published in the related works] and that the extended SSH model is not very convincing in terms of novelty.

We thank the reviewer for their time and constructive feedback that enabled us to significantly improve the manuscript.

Reviewer 2:

The author has already answered the question, resolved my doubts, and I agree to publish this article.

We thank the reviewer for their time and constructive feedback that enabled us to significantly improve the manuscript.

Reviewer 3:

The changes the authors made have improved the manuscript and the response has addressed my previous concerns. I would like to provide my recommendation for its publication.

We thank the reviewer for their time and constructive feedback that enabled us to significantly improve the manuscript.